# A Novel Decentralized Blockchain Architecture for the Preservation of Privacy and Data Security against Cyberattacks in Healthcare

**DOI:** 10.3390/s22155921

**Published:** 2022-08-08

**Authors:** Ajitesh Kumar, Akhilesh Kumar Singh, Ijaz Ahmad, Pradeep Kumar Singh, Pawan Kumar Verma, Khalid A. Alissa, Mohit Bajaj, Ateeq Ur Rehman, Elsayed Tag-Eldin

**Affiliations:** 1Department of Computer Engineering and Applications, GLA University, Mathura 281406, India; ajitesh.kumar@gla.ac.in (A.K.); akhileshkr.singh@gla.ac.in (A.K.S.); pradeepsingh.gla@gla.ac.in (P.K.S.); anushree.gla@gla.ac.in (A.); 2Institute of Computer Sciences and Information Technology (ICS/IT), The University of Agriculture, Peshawar 25130, Pakistan; 3Department of Computer Science Engineering, MIT Art, Design and Technology University, Pune 412201, India; pawankumar.verma@mituniversity.edu.in; 4SAUDI ARAMCO Cybersecurity Chair, Networks and Communications Department, College of Computer Science and Information Technology, Imam Abdulrahman Bin Faisal University, P.O. Box 1982, Dammam 31441, Saudi Arabia; kaalissa@iau.edu.sa; 5Department of Electrical and Electronics Engineering, National Institute of Technology Delhi, Delhi 110040, India; mohitbajaj@nitdelhi.ac.in; 6Department of Electrical Engineering, Graphic Era (Deemed to be University), Dehradun 248002, India; 7College of Internet of Things Engineering, Hohai University, Changzhou 213022, China; 8Faculty of Engineering and Technology, Future University in Egypt, New Cairo 11835, Egypt

**Keywords:** blockchain, healthcare, cyber security, cyberattack

## Abstract

Nowadays, in a world full of uncertainties and the threat of digital and cyber-attacks, blockchain technology is one of the major critical developments playing a vital role in the creative professional world. Along with energy, finance, governance, etc., the healthcare sector is one of the most prominent areas where blockchain technology is being used. We all are aware that data constitute our wealth and our currency; vulnerability and security become even more significant and a vital point of concern for healthcare. Recent cyberattacks have raised the questions of planning, requirement, and implementation to develop more cyber-secure models. This paper is based on a blockchain that classifies network participants into clusters and preserves a single copy of the blockchain for every cluster. The paper introduces a novel blockchain mechanism for secure healthcare sector data management, which reduces the communicational and computational overhead costs compared to the existing bitcoin network and the lightweight blockchain architecture. The paper also discusses how the proposed design can be utilized to address the recognized threats. The experimental results show that, as the number of nodes rises, the suggested architecture speeds up ledger updates by 63% and reduces network traffic by 10 times.

## 1. Introduction

Industry plays a major role in the development of any country. In the modern scenario, the healthcare industry is gaining much in popularity and is considered the most prominent sector. The most important aspect of the digital healthcare sector in the modern era is data. Due to the increase in cyber threats and ransomware, the privacy and security of data are of much concern. A look at traditional times will reveal that cyber-crime emerged in the late 1970s, from where it began to spam into malware and viruses; cyber-criminals continue to be on the lookout for areas they can attack and access important personal credentials. A recent attractive target of such cyber-crime attackers to steal individuals’ personal and financial credentials is the healthcare sector. It is prone to such attacks and lags in security aspects compared to other industries. Various regulatory legislations are moving in this direction, including the Health Information Technology for Economic and Clinical Health Act (HITECH), the Patient Protection and Affordable Care Act (ACA), and the Health Insurance Portability and Accountability Act of 1996 (HIPAA). Researchers in the blockchain are proposing several measures to secure healthcare sector data. Blockchain is an emerging technology used to create and preserve data and provide security and a distributed database for transactions (Figure 1). Blockchain technology provides greater transparency and better protection and increases efficiency with improved traceability.

In blockchain technology, implementation is carried out via N blocks, in which each and every successor block is securely linked in a chain with the predecessor block. The block contains the data, the hash generated by the predecessor, and the hash generated for the current node. If the data of any current block is altered, then the hash of all the subsequent blocks is changed and hence will be incorrect. This shows the distributed decentralized working of blockchain, where every node is considered a participant of the chain and participates in maintaining the consistency of the blockchain, thereby maintaining the integrity of the data.

Similarly, nowadays, blockchain is getting a lot of consideration and possesses the ability to revolutionize the existing structure of technologies connected through the Internet. It is a time-stamped series of immutable records of data and a distributed, decentralized, public ledger that is managed by a group of computers rather than being owned by any single organization. Hence, the fusion of such two admired and useful technologies can be a boon to several industries and organizations.

The healthcare industry is a prominent sector for the application of blockchain technology, and it is energizing revolutionary changes in traditional diagnostic methods to modern diagnosis systems based on patients’ history-track. Consider a scenario where before prescribing treatment to the patient, the doctor can look for past medical history or treatment and medicines prescribed from other hospitals for the patient. In such a case, the best treatment can be decided and prescribed. Moreover, the national economy also can be boosted by such a revolutionary change. User trust is a prominent and important factor in sharing such vital information. Any lapse in sharing will generate distrust among patients for e-healthcare systems. In a secure healthcare system there exists a plethora of work, but some concerns, such as increased return traffic, moderate handling of recognized threats, etc., provide future research directions. In such a scenario, blockchain technology in healthcare can be the more favorable approach to attain security from recognized threats and mitigate the increased traffic rate.

By using blockchain as a secure mechanism to protect e-healthcare data, several compelling advantages and features can be deployed without the mistrust or involvement of third parties. Blockchain is also considered tamper-proof, highly reliable, and always available due to its consensus mechanism, unique data structures, and replications. With its numerous benefits and advantages, the market is now displaying an increased profit shift by using blockchain and improved blockchain market awareness. Figure 2 shows the investment of healthcare industries in the blockchain [1].

Figure 3 shows the spread of awareness for blockchain mechanisms among different categories of persons associated with e-healthcare [1]. But even with the high spread of awareness, there are still many barriers, hurdles, and challenges restricting the adoption of blockchain technologies, as depicted by Figure 4 [1].

Since the blockchain acts as a decentralized distributed architecture without the involvement of any central authority, the records and data will be public and can be verified easily. When applied to the healthcare sector, the data blocks are not fully public in nature but can be accessed, retrieved, or verified by authorized personnel before accepting them as part of the blockchain. Every block of record in the blockchain is attached with a timestamp and a cryptographic digital signature to preserve the authenticity of the blockchain. To access a blockchain record, the node’s address serves as the public key and the credential password serves as a private key.

This technique allows the record and data stored in the blockchain node to be incorruptible and allows the source of origin to trace it, keeping the source information hidden.

Some key application areas of the healthcare sector where the blockchain plays a prominent role in securing data include electronic medical records (EMR), electronic health records (EHR), remote patient monitoring, the pharmaceuticals supply chain, and health insurance claims. Although electronic medical record (EMR) and electronic health record (EHR) are often used as synonyms, there is a slight difference between the two terms. An EMR indicates the treatment and history of a patient in single practice and is considered to be the digitized version of the patient chart and prescription. In contrast, an EHR affords a wider and broader view and focuses on a patient’s health in its totality. Some applications that integrate and support blockchain technology in the field of EMR and EHR are MedRec [3], FHIRChain [9], Ethereum applications, MedBlock [10], BlockHIE [11], and Ancile [12]. Remote patient monitoring deals with monitoring and supervising patient health by means of electronic media. It collects and reports various patient health parameters via mobile devices, including devices such as the Internet of Things and body sensor networks. The blockchain proves to be a major tool for collecting, accessing, and retrieving data in this field. Various applications have been proposed in this area, which includes hyperledger fabric technology [13], Ethereum smart contracts [7], and IOBHealth [14].

Another application area in the healthcare sector where blockchain plays a major role is the pharmaceuticals supply chain. This area can restrict the patient from having, consuming, or delivering inadequate or counterfeit medicines that can create critical or ill consequences in the patient’s health. Various blockchain technologies and advancements have been proposed in this area and include Modum.io AG [15], blockchain-based drug standardization, falsified drug detection using blockchain, and secure traceable pharmaceuticals supply chain technique using blockchain. The other field in the healthcare sector that can benefit by using blockchain is health insurance claims. In this area, blockchain provides transparency, immutability, and auditability of the data stored related to insurance claims. Plenty of potential and research are evident in this area. MiStore [16] is considered a medical and health insurance system using blockchain that secures and encrypts health-related data [1,9,16,17,18,19,20]. The present research article discusses and reviews the work currently being carried out in the field of healthcare blockchain applications along with its challenges, open issues in the existing scenario, future investigations of the blockchain in the healthcare sector, and a research perspective for securing healthcare data by presenting a novel approach verified by results and comparison with the existing approach. The following are some key contributions of our article;

It proposes a decentralized blockchain architecture for privacy and data protection against healthcare cyber-attacks.Using blockchain, we are proposing a privacy-preserving and secure hash algorithm.The effectiveness of our proposed framework is determined by performing a security review on cyber-attack.The performance of our proposed novel decentralized blockchain architecture is evaluated by comparison with the existing models with respect to several performance criteria.

The remainder of this article is as follows. Section 2 details the literature review that has been taken into consideration for the current article. Section 3 proposes a novel blockchain mechanism for securing a healthcare sector data management architecture that reduces communicational and computational overhead costs compared to the bitcoin network. A lightweight blockchain architecture follows its result, comparison, and discussion in Section 4, followed by a conclusion and references.

## 2. Related Work

Currently, various review research articles in the literature deal with the application of blockchain technology in the area of the Internet of Things [2], finance [6], government [21], the energy sector [4,22], and security and privacy [8,23]. Some researchers have focused on the area of the Internet of medical things and healthcare using blockchain technology. Numerous researchers have shown concern and addressed medical health records and data-sharing issues in recent times. Reference [24] proposed a completely functional and working prototype implemented via blockchain known as MedRec that supervises medical health records. References [15,25] mentioned the limitations and issues faced in adopting blockchain in electronic health record systems, including performance and scalability issues, secure and authorized identification, lack of incentive, and over-and under- usability. Reference [5] presented a survey in data management for patients that searches the self-sovereign aspect using blockchain. Reference [26] reviewed the healthcare sector using a blockchain approach from the highest-layer architecture to the lowest-layer algorithms. Reference [27] reviewed and performed a survey on blockchain’s application area in the healthcare sector, with a primary focus on SWOT analysis. A comprehensive framework for IoT devices has been proposed, based on updated blockchain models [28]. In [29], the authors have suggested a wise use of IoT and blockchain merger in the field of an integrated autonomous sewage management system. The framework and the model can be viewed and referenced in many of the current approaches to the blockchain.

The strengths of the private key, public key, and blockchain are combined in this hybrid model. Legacy systems play an important role in the healthcare sector since, in general, they internally exchange health and medical reports [30] and generally do not operate with external systems [31]. Several benefits [14] can be observed by combining these two paradigms, including improved and integrated healthcare services. For a cyber-physical system, a privacy-preserving scheme has been proposed for advanced obstinate threads [32,33,34,35]. Here the authors check the level of data privacy preserved by calculating the differences between original and transferred data using an index level of privacy.

Holochain-based security is a confidentiality framework that solves the difficulty of resource constrain in IoT healthcare systems [36]. Here the authors discussed the fact that the order of time complexity in a blockchain network [37] is increasing exponentially with several nodes, but in the case of holochain, the average order of time complexity remains constant even while the number of connected nodes increases. 

Data sharing between organizations proves to be another vital challenge [38,39] that demands the health and medical dataset created by one healthcare personnel be shared securely with other healthcare-requesting entities such as a doctor or a healthcare research entity. The summary of various applications in the field of healthcare using the blockchain is summarized in Table 1.

### Research Gap in Current Solution and Motivation

Healthcare data would become an especially alluring target for cyber criminals since it is held in centralized databases in layers, which is a major flaw in the existing system. Numerous studies have demonstrated that centralization raises cyber threats and necessitates faith in a central body. Additionally, if the EHR is removed from the patient’s database, the record may be destroyed forever, which is another problem that needs to be carefully addressed. So, it is crucial that we find an alternative system that is only accessible by authorized stakeholders. With the outsourcing of private medical data stored in the cloud, the privacy of EMRs is experiencing a major challenge because there is a risk of patient data being disclosed to unauthorized users. In actuality, in the current solution, most EMRs are kept in centralized databases, which raises the security risk perimeter and necessitates a reliance on single authorities that is ineffective at safeguarding data from insider threats. That would be the idea behind this suggested solution.

## 3. The Proposed Approach

First, we used bitcoin’s decentralized currency which is underpinned via a peer-to-peer network system. In this, when any transaction happens, it is known to the entire network, and all the miners who receive this transaction can verify or check it through their signatures which are in the transaction. Further, these miners append this verified transaction into their own block. This shows that the blockchain uses multiple miners to verify a single transaction; this shows its resolution. The data flow diagram of the proposed architecture is shown in Figure 5a. and the proposed architecture for blockchain-based secure healthcare data management is shown in Figure 5b. It depicts the main architectural components and their functions and relations. Following that, we will go over various components of the proposed architecture.

### 3.1. Participant Nodes

We used analytical calculations for the sensor with concentric electrodes to determine the optimal geometric dimensions, which were conducted in [35]. Using the mathematical relationship from and Equation (1), the sensor response function was calculated as:

All the entities chosen as network participants show equal influence against the network and exchange a copy of the activity data as a ledger in a decentralized design as shown in Figure 5b. The blockchain allows for the direct exchange of a database via a distributed ledger without the use of an intermediary. Network nodes are in charge of processing and storing the transactions. After an agreement has been established, the nodes update the ledger.

The blockchain network’s transaction data are duplicated on the network nodes as a series of all transactions joined and linked with each other, moving backward toward the first transaction. Infrastructure changes are transparent and secure since they are immediately visible.

Blocks are used to store transactions in the blockchain. A cryptographic secure hash function links each block of the series chain to the predecessor block. Any effort to change the text of a block will have an impact on the blocks that follow it in the series chain. As a result, a malicious harmful attacker needs to change each and every subsequent block in the chain computationally to update a single block. Because the chained blocks’ copy is copied over numerous nodes, this becomes very difficult to modify. This makes the proposed architecture more secure.

### 3.2. Secure Hashing Encryption

All the transactions in an individual block in the blockchain are processed when a specified count of blocks is concatenated to the longest series chain to handle transactions in the proposed architecture. So, in this section, we propose a secure encryption algorithm to generate a blockchain. We discuss the proposed algorithm is below (Algorithm 1), and in the next section we show the performance evaluation of the proposed work.


**Algorithm 1 **Proposed Algorithm
**Input:** In the blockchain, Input “data.” Data can be anything like a word, a line, a paragraph, or multiple paragraphs.data can be transactions of any digital currency.       
{dataevent [2256−1]}
**Output:** A block created for every set of data. And as a result, a chain of the block is created known as the blockchain.

{ Hash Value of medical input data }


$$in\;the\;interval\;of\;positive\;integer\;\left[0,\;2^{256}\;\right)$$

**Assumption:** Blockchain is a giant public ledger that tracks all bitcoin transactions ever made. These incoming transactions are referred to as “inputs. “Transactions entered in the order in which they occurred.START: FOR EVERY 
{dataevent [2256−1]}
Step 1: Pre-processing: Generate a hash for the data. The hash depends upon the previous hash and the data in the current hash. Initial hash value (ph = 2,3,5,7,11,13,17,19)

$$int\;\left((\;\sqrt{p}\;mod\;1\right)*16^{8})$$

Step 2: Hash Computing Using Preprocessing Data:Now, we compute additional 48 words, totaling the number of words to 64. Here we compute the 10th word (
$W_{9}$
).

$$W_{t}=ROTL\;(\sigma_{1}\left(W_{t-2}\right)+W_{t-7}+\sigma_{0}\left(W_{t-8}\right)+W_{t-9})$$


$$\sigma_{0}=ROTL^{9\;}\left(m1\right)\;\vee \;ROTL^{12}\left(m1\right)\vee SHL^{3}\left(m1\right)$$


$$\sigma_{1}=ROTL^{11\;}\left(m1\right)\;\vee \;ROTL^{13}\left(m1\right)\vee SHL^{34}\left(m1\right)$$


$$t=10\;\mathrm{and}\;m_{1}=\;W_{t-9}$$


ROTL−Left Rotate

Step 3: A small change in data creates a new hash known as AVALANCHE EFFECT. Create a block for a particular set of data. A block is accepted when 51% of the nodes verify the data within the block. That is known as BYZANTINE FAULT TOLERANCE.Step 4: Generate a new block with hash values (a, b, c, d, e, f, g, h) for other sets of data.

For t=0 to 63

$$T_{1}=h+\overset{256}{\underset{1}{\sum }}e+\mathrm{ch}\left(e,f,g\right)+\;k_{t}^{256}+W_{t}$$

$$\;T_{2}=h+\overset{256}{\underset{1}{\sum }}a+\mathrm{maj}\left(a,b,c\right)$$

h=g=f

$$f=d+\;T_{1}$$

$$e=f+T_{1}$$

d=c

c=b=a 

$$and\;a=\;T_{1\;}+\;T_{2}$$
Step 5: The chain of the block created is known as the blockchain.Step 6: TO COMPUTE THE FINAL HASH, RUN THE 64 ITERATIONS OF STEP 4.

A cryptographic hash function converts a message string of a given length into an alphanumeric string of the specific length. In the proposed algorithm, the length of the input data has a limit of 2^256^ − 1 and the output will always be 256 bits in the length. In the first step, we begin with a hash preprocess via converting the message in the binary form. Prior to computing the blocks, we set the initial hash value using the given formula in step 1. Steps 2 and 3 are allied hash computing phase. For computing the final hash, we run 64 iterations of step 4.

## 4. Experimental Setup and Implementation

We created a blockchain structure that is relatively similar to the actual blockchain used by bitcoin in the implementation phase. This demonstrates that the blocks are nothing more than user-generated transaction data. Upon validation, these blocks are added to the blockchain. Following are the stages of a bitcoin-like blockchain’s execution along with the proposed experimental setup which is shown in Table 2.

### 4.1. Creating a Blockchain Class

We create a blockchain class that is a blueprint of our blockchain argument. We initialize the chain, which is nothing but an empty list. We use a list because the list is a mutable data type, so we can easily add new blocks to the list (chain). We call this the “create blockchain” method, which takes three arguments—the object itself (self), the proof of work, and the previous block’s hash. In other words, we initialize the first block (genesis block) of the blockchain and add it to the chain. The “create blockchain” method is responsible for adding the block (which is mined) to our blockchain as well as returning the block.

### 4.2. Proof of Work

This method returns the last block of the blockchain and is the most interesting part of the blockchain. The proof-of-work method finds the appropriate proof of work. So, in order to mine the block, miners must solve this algorithm.

The above algorithm is simple and can be easily solved, but to make it more complex we can increase the number of leading zeros and increase the complexity of the hash problem. Now the proof of work is verified.

### 4.3. Hash Algorithm: Block Passed in the Method

This method functions as the security of our blockchain, as it checks whether the cryptographic link is still intact and the proof of work of the block meets the requirement. So we can say that it is the guarantor of the safety of our blockchain. It comes out True if everything is okay; otherwise, it comes out False.

### 4.4. Creating Complete Blockchain

Next we create the class “blockchain” object, which is our real blockchain (until now we are have been working only with a blueprint). Create an instance of flask class which is named as an app. [app = Flask(__name__)] This block of code is responsible for mining and displaying information regarding the newly mined block whenever we use a GET- type request in Postman to call the mine block function. Here, 200 is the http code for the task being carried out. This block of code display completes the blockchain in our postman program.

### 4.5. Performance Analysis

As the number of blocks continuously increases, traffic and computational overhead increase. Figure 6 displays the quantity of data sent for updating the ledger individually at each node in the secure network and each secure medical data center in the proposed novel architecture.

The quantity of data used by the bitcoin network increases linearly in all circumstances, resulting in increased traffic and incurred computational overheads. In general, on average, the cumulative amount of data transmitted using the proposed novel approach is 13 times lower than in the traditional bitcoin network and 3 times lower than in the LW blockchain technology.

Figure 7 depicts the total processing time and restoring time for updating the ledger individually at each node in the secure network along with that at each secure medical data center in the proposed architecture. The cumulative sum of the data transferred and exchanged is 10 times lower than in the traditional bitcoin network and LW blockchain against an increase in the number of nodes. The security and privacy accuracy are used to evaluate the effectiveness of our proposed decentralized architecture using a secure hashing algorithm on sensitive health care records. Figure 8 and Figure 9 show the experimental results, where our proposed algorithm ensures high accuracy in comparison with bitcoin and LW blockchain.

Furthermore, compared with the traditional bitcoin network, the proposed novel architecture achieves a 63% reduction in total processing time and restoring time against the replication of data and updating of the ledger. The findings obtained with the proposed secure architecture show its efficacy and appeal for prospective adoption by medical organizations looking to leverage blockchain technology for healthcare data management.

We then looked at how the network throughput performed in varying combinations of the network participants in Figure 10. We noticed that the majority of the scheduling nodes had a significant impact on operational throughput but delegates, contributors, and administrators had a minor impact due to network latency. We also discovered that with the lowest load, up to 20 transactions could be completed.

The impact of the number of cycles as well as the inaccuracy rate is seen in Figure 11. In Figure 12, we observed the overall perforce of the proposed framework and algorithm via finding the better response time when the number of records increased while maintaining the balance of bandwidth between shared nodes.

## 5. Conclusions

In this study we were able successfully to propose a novel blockchain mechanism for secure healthcare sector data management; we also discussed how the proposed design tackles recognized threats. The benefits of the proposed design include improved security against known threats, a slower rate of traffic growth, more transparency, instantaneous traceability, robustness (replicating the data in various locations), etc. Experimental simulated results indicate that when there is an increase in the number of nodes, our architecture speeds up ledger updates by 63% and reduces network traffic by 10 times. This research experience shows that improvement and gradual slow integration or migration of healthcare sector information sharing will definitely serve the stated purpose and will surely protect the interests of all parties. Also, considering the cumulative amount of network data traffic created and the time taken to process secure data, the proposed architecture shows improvement over the traditional bitcoin network and the lightweight network. Furthermore, storing a large amount of data may create inefficiency and lead to more expensive issues in the proposed design. Therefore, the aforementioned issues can be the future research direction for this topic.

## Figures and Tables

**Figure 1 sensors-22-05921-f001:**
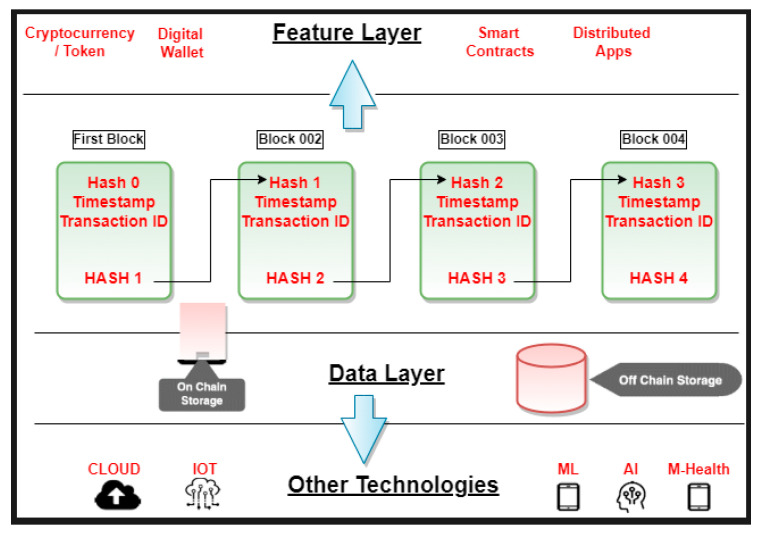
Blockchain in its simplest form.

**Figure 2 sensors-22-05921-f002:**
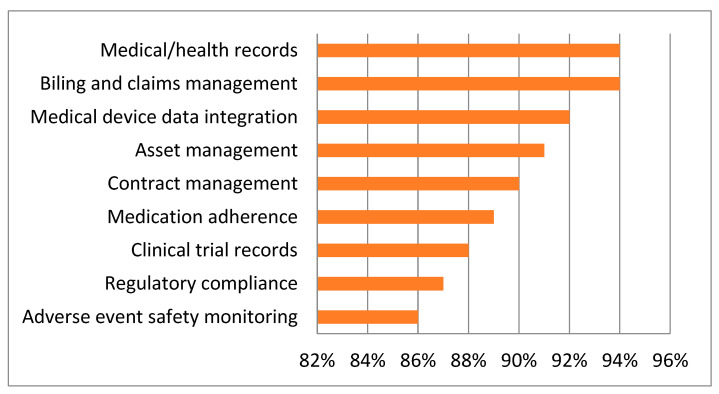
Investment of healthcare industries in the blockchain [2,3].

**Figure 3 sensors-22-05921-f003:**
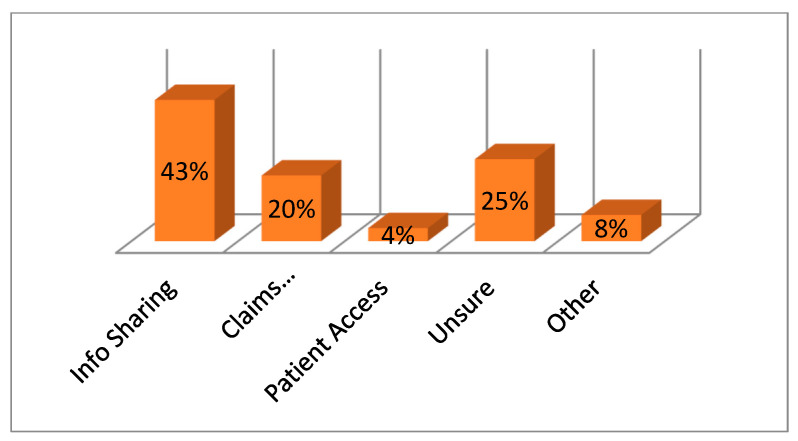
Blockchain awareness among e-healthcare persons [4,5].

**Figure 4 sensors-22-05921-f004:**
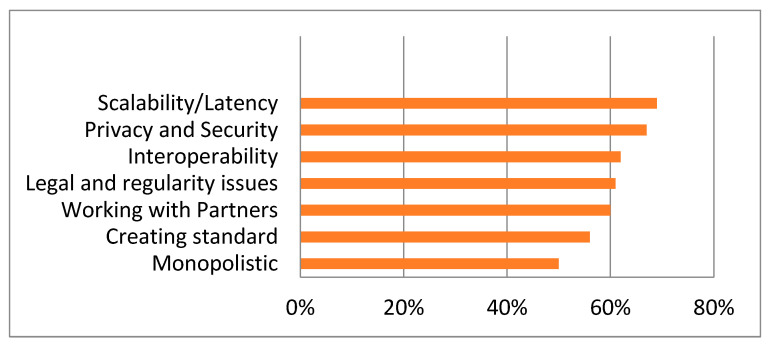
Challenges and barriers in the adoption of blockchain technology [6,7,8].

**Figure 5 sensors-22-05921-f005:**
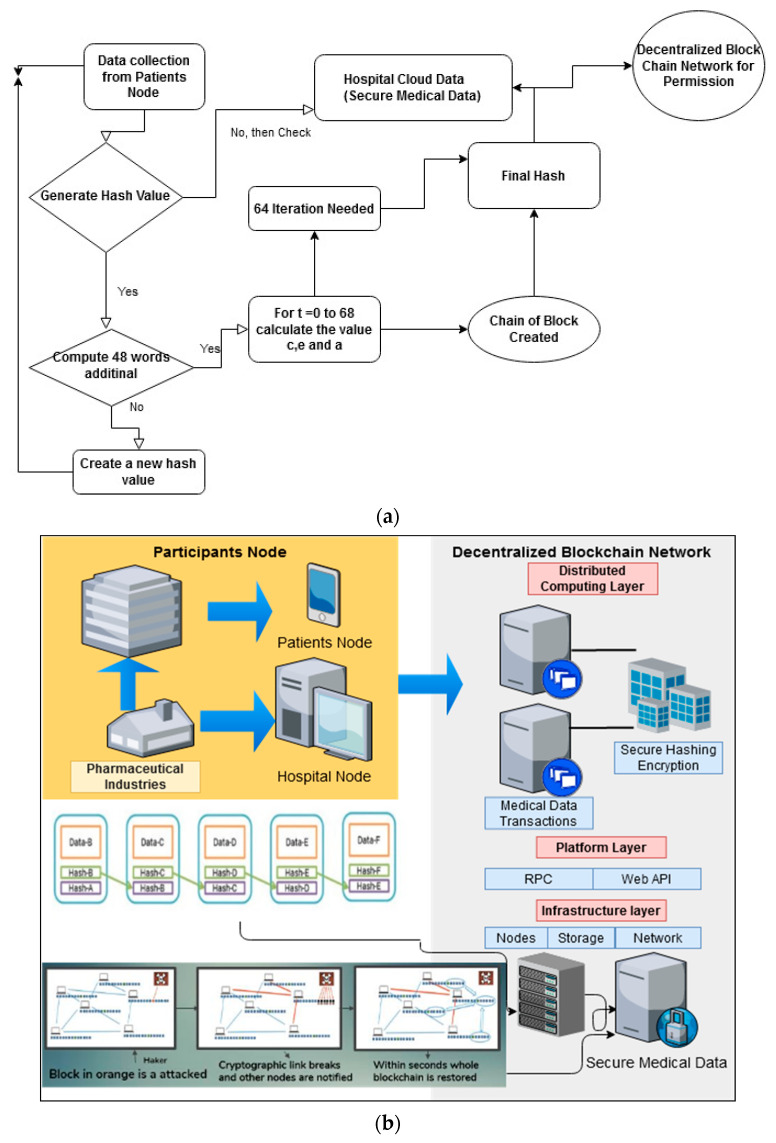
(**a**) Data Flow Diagram of the Proposed Approach. (**b**) The proposed secure architecture for blockchain-based healthcare data management.

**Figure 6 sensors-22-05921-f006:**
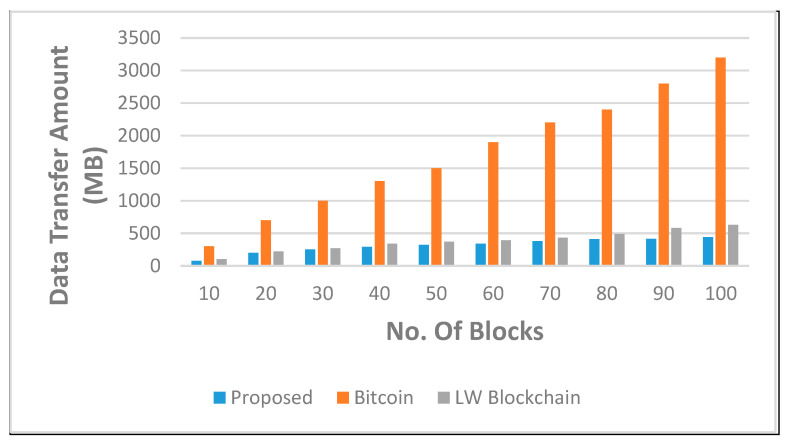
The count of blocks grows larger against the aggregate of data moved via bitcoin, LW blockchain, and our proposed secure blockchain architecture.

**Figure 7 sensors-22-05921-f007:**
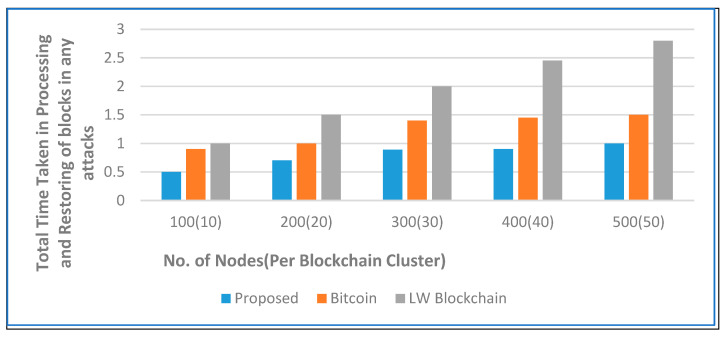
Total processing time and restoring time of blocks against the increase in the number of nodes.

**Figure 8 sensors-22-05921-f008:**
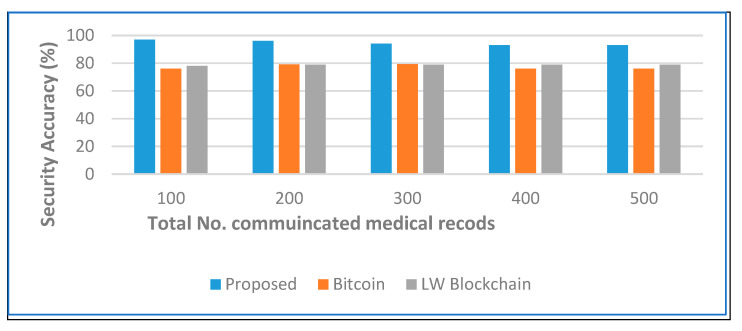
Security accuracy analysis when the number of medical records increases.

**Figure 9 sensors-22-05921-f009:**
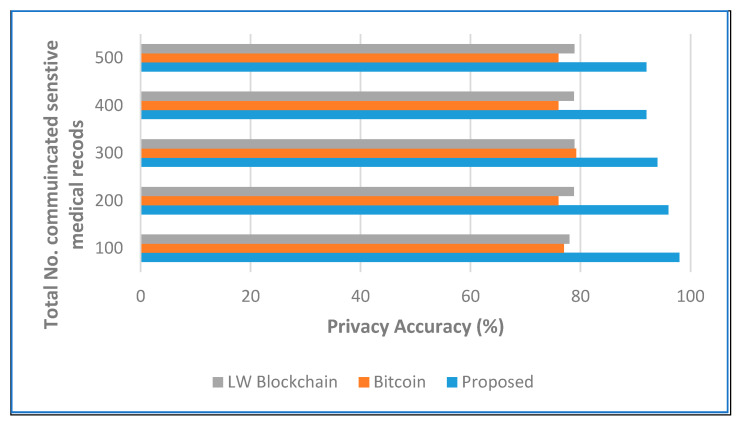
Privacy accuracy analysis when the number of medical records increases.

**Figure 10 sensors-22-05921-f010:**
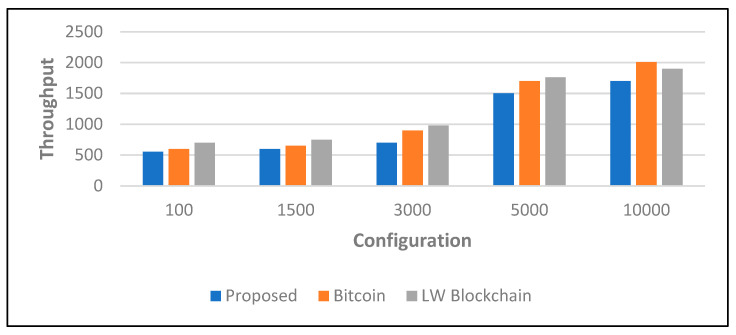
Analysis of transaction throughput.

**Figure 11 sensors-22-05921-f011:**
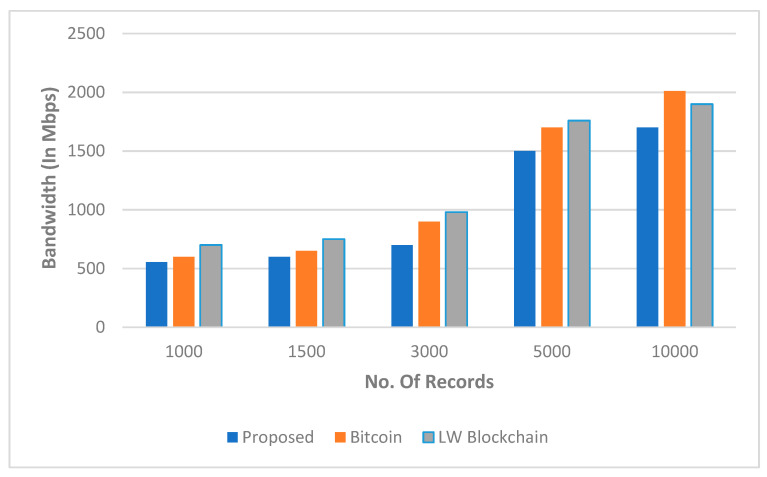
The simulated outcomes according to the bandwidth and number of iterations.

**Figure 12 sensors-22-05921-f012:**
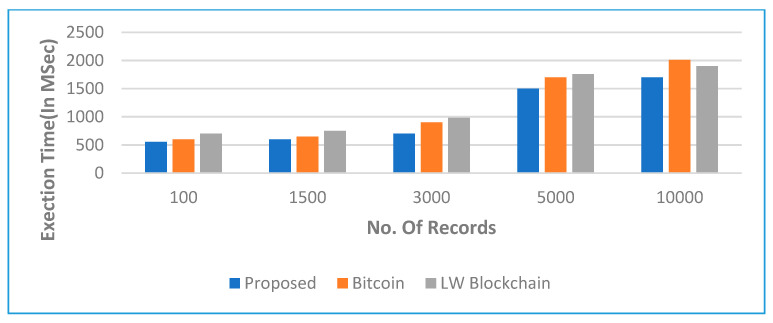
According to the execution time and number of medical records, the simulated outcomes.

**Table 1 sensors-22-05921-t001:** Contributions of various researchers in the healthcare sector using the blockchain.

Application Area	Summary of Work Carried Out or Proposed
	Ekblaw, A. et al. [3] suggested the development of a patient-centered approach for bias-free, non-hidden, authorized access and view of patient medical history (Ethereum model proposed/used)
	Xia, Q. et al. [40] applied techniques on cloud databases for sharing medical data. (Ethereum model proposed/used)
	Zhang P et al. [41]: An EMR application using the blockchain is proposed (Ethereum model proposed/used)
	Fan, K. et al. [10]: An EMR management application using the blockchain is proposed. (Proprietary model proposed/used)
	Jiang, S. et al. [11] proposed a system for storage of off chain data and verification of on chain in healthcare sector. (Proprietary model proposed/used).
	Dagher, G.G. et al. [12]: A secured EHR system is proposed that secures personal information related to patients’ health. (Ethereum model proposed/used)
Remote patient monitoring (RPM)	Ichikawa, D. et al. [13] suggested a mobile health blockchain-based system for insomnia. (Hyperledger fabric model proposed/used)
	Griggs, K.N. et al. [7] migrated the use of the IoT with blockchain technology for data collection and security.
Pharmaceutical supply chain (PSC)	Bocek, T. et al. [15]. proposed a system for maintaining the drug temperature record while in transit and transportation. (Ethereum model proposed/used)
	Bocek, T. et al. [15] suggested a mechanism to explore a blockchain implementation in healthcare for efficient traceability to the drugs supply.
	Miler et al. [30] proposed a design on immutable and secure PSC. (Hyperledger fabric model proposed/used)
Health insurance claims (HIC)	Zaman et al. [36] suggested work and implementation on the medical and health insurance claims storage mechanism. (Ethereum model proposed/used)

**Table 2 sensors-22-05921-t002:** Experimental setup.

Hardware Setup	Processor	Intel Xeon X5355 2.66 GHz (8 Cores)
	Memory	8 GB RAM
	Storage	2 × 146 GM
	Network Interface	1 GbE
Software Setup	Anaconda (Spyder)	IDE Used for Implementation
	Flask	A web framework for blockchain communication
	Postman	Display the information
Simulation Criteria	Block Size	1 MB
	Hash Function Size	256 bytes
	Nodes (Blockchain cluster)	100(10), 200(20), 300(30), and 400(40)

## Data Availability

The data presented in this study are available upon request from the corresponding author.

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
