# Peer review of "A Novel Decentralized Blockchain Architecture for the Preservation of Privacy and Data Security against Cyberattacks in Healthcare"

_sensors, 2022, doi:10.3390/s22155921_

Round 1

Reviewer 1 Report

Comments and Suggestions for Authors

1) In introduction, introduce the problem, motivate the problem, and summarize the contributions

2) Related work: There are many published works and authors are suggested to do a detailed literature survey and in addition mention the disadvatages of the exsisting works and how the proposed work overcomes these limitations. Some works to include in related works are

https://fanyv88.com:443/https/link.springer.com/chapter/10.1007/978-3-030-76216-2_14  
https://fanyv88.com:443/https/link.springer.com/chapter/10.1007/978-3-030-76216-2_3
https://fanyv88.com:443/https/ieeexplore.ieee.org/document/8985278
https://fanyv88.com:443/https/ieeexplore.ieee.org/document/9377310

3) Performance of the proposed method should be comapred with the exsisting 3 related works

4) Include the proposed approach advatages and limitations

5) Robustness of the proposed study can be included

Author Response

Thank you for valuable comments.

Reviewer 2 Report

Comments and Suggestions for Authors

The proposal is appealing and interesting, and the method deserves some consideration. Moreover, the paper is almost well written and well organized.

- The text, in general, reads well, but a grammatical revision could improve it further.

- The paper adequately puts the progress it reports in the context of previous work, representative referencing and first discussion.

Author Response

Thanks for valuable comments.

Reviewer 3 Report

Comments and Suggestions for Authors

Author Response

Thanks for valuable comments.

Round 2

Reviewer 3 Report

Comments and Suggestions for Authors

Present form is publishable